# Acceptance, Advocacy, and Perception of Health Care Providers on COVID-19 Vaccine: Comparing Early Stage of COVID-19 Vaccination with Latter Stage in the Eastern Region of Saudi Arabia

**DOI:** 10.3390/vaccines11020488

**Published:** 2023-02-20

**Authors:** Eman M. Almusalami, Mohammed I. Al-Bazroun, Amal I. Alhasawi, Fatimah S. Alahmed, Zahra M. Al-Muslim, Lubana I. Al-Bazroun, Maryam Muslim, Chandni Saha, Elbert Kay, Zeyad A. Alzahrani, Gasmelseed Y. Ahmed, Abbas Al Mutair

**Affiliations:** 1Research Center, Almoosa Specialist Hospital, Al-Ahsa 36342, Saudi Arabia; 2Qatif Health Network, Qatif 32654, Saudi Arabia; 3Dhahran Eye Specialist Hospital, Dhahran 34257, Saudi Arabia; 4Almoosa College of Health Sciences, Al-Ahsa 36342, Saudi Arabia; 5Population Health Department, John Hopkins Aramco Healthcare, Al-khober 31952, Saudi Arabia; 6Administration, Presidency of State Security Hospital, Riyadh 12223, Saudi Arabia; 7Columbia University Hospital, New York, NY 10027-6907, USA; 8Faculty of Medicine and Health Sciences, Managil University for Sciences and Technology, Managil 21111, Sudan; 9School of Nursing, University of Wollongong, Wollongong 2522, Australia; 10Nursing Department, Prince Sultan Military College, Dhahran 34313, Saudi Arabia; 11Medical-Surgical Nursing Department, Princess Nourah Bint Abdulrahman University, Riyadh 84428, Saudi Arabia

**Keywords:** COVID-19, vaccine, HCPs, acceptance, advocacy

## Abstract

Vaccination of healthcare providers has recently gained focused attention of public health officials. As HCPs have direct contact with the population, and HCPs significantly influence the population, this study aimed to compare the acceptance rate, advocacy rate, and beliefs about the COVID-19 vaccine among HCPs in two time periods. In this repeated cross-sectional study, different HCPs were assessed in two periods ten months apart, i.e., November to December 2020 and September to October 2021, which were before and after COVID-19 vaccine approval by authorities. The study was conducted in Qatif Central Hospital, Eastern Region of Saudi Arabia. There were 609 respondents: 236 participants in the first period and 373 participants in the second period. Only 13 participants did not get the COVID-19 vaccine. There was around a 40% difference in the acceptance rate between the two study periods; the latter period was higher at 94.7%. Furthermore, 24.1% was the difference between the willingness to advocate the COVID-19 vaccine for others; the first period had a lower percentage (60.1%). Overall, results of the study showed that vaccine hesitancy, as well as the willingness to advocate for the vaccine, were improved between the pre-vaccine approval period and post-vaccine approval period, showing that the efforts made by the government improved COVID-19 acceptance and advocacy among HCPs. However, vaccine hesitancy is not a new issue, and for a better understanding of HCPs’ beliefs, a qualitative study is needed.

## 1. Background

The first case of SARS-CoV-2 infection in Saudi Arabia was identified in March 2020 [1,2,3,4]. Since then, this novel virus’s infection cases have rapidly and significantly increased. Eventually, it was declared a global pandemic by the World Health Organization (WHO) in March 2020 [5]. The speed of disease transmission hindered rigorous research evidence regarding COVID-19 disease prognosis [6,7,8]. This unexpectedly resulted in increasing COVID-19 infection disease mortality [6,7,8]. Although several extensive efforts were made to explore treatment options, there has been no proven effective treatment, except preventive measures like vaccines [7,8,9]. However, some available drugs were repurposed to be used for COVID-19, and they showed positive results in mortality.

Vaccinations are among the most important tools and preventive measures for reducing the spread of infectious diseases. The WHO estimated that vaccines prevented at least 10 million deaths between 2010 and 2015 worldwide [10,11]. Vaccinations as a preventive measure are considered one of the most successful public health interventions. Hence, the developed COVID-19 vaccines represent an effective weapon in facing and defeating the pandemic [12,13].

Despite the challenges and struggles associated with managing COVID-19 waves and variants, the vaccines that were approved for use have created a window to fight this COVID-19 pandemic effectively through achieving herd immunity. The rates of vaccination coverage vary considerably between countries [14]. Saudi Arabia was one of the first countries to implement a COVID-19 vaccination program. The Saudi Ministry of Health provided both Pfizer-BioNTech and AstraZeneca vaccines for both citizens and residents [15]. The vaccination campaign in KSA was started on 17 December 2020 with the Pfizer-BioNTech vaccine and later the AstraZeneca vaccine in February 2021 [15].

The repeated waves and emergence of new strains of the COVID-19 virus are expected to continue for unanticipated periods, resulting in enormous burdens of morbidity and mortality. Several newly developed vaccines against COVID-19 are currently available and accessible. The COVID-19 pandemic provides the opportunity for the public health community to build vaccine literacy knowledge and confidence, thereby supporting the uptake of the COVID-19 vaccine specifically and for overall immunization vaccination programs for all vaccine-preventable infectious diseases. Multiple factors have the potential to influence the acceptance rates of vaccines. A global study including 13,426 people from 19 different countries showed that 71.5% of the study participants were likely to receive the COVID-19 vaccine, and 48.1% of the participants were open to accepting their employer’s recommendation of receiving the COVID-19 vaccine [16]. Additionally, COVID-19 vaccine acceptance rates were different among the 19 countries, ranging from almost 90% in China to around 55% in Russia [16].

Vaccine hesitancy is defined by the SAGE Working Group as a “delay in acceptance or refusal of vaccination despite the availability of vaccination services” [17]. It is one of the major growing public health concerns, and it is an emerging public health challenge issue. It is reinforced by false information related to vaccine effectiveness and safety [18,19,20]. Vaccine hesitancy was ranked in the top ten threats to global health [21]. Understanding factors that influence vaccination helps immunization programmers to monitor, change, check, and assess strategies to improve and support vaccination acceptance. Other influential factors such as media monitoring and collaboration with partners and stakeholders can help in vaccine acceptance. In addition, community- and school-based programs are inspired approaches to reduce vaccine hesitancy, increase acceptance rates, and thus result in a good response to vaccination crises and recovery. The community should be well informed about new insights regarding demand during and after outbreaks such as the COVID-19 pandemic [22].

Vaccine decliners in some healthcare sectors must sign statements admitting the risk they are undertaking for themselves and their patients. These efforts caused an increase in voluntary acceptance of the vaccine [23]. In addition, to increase the willingness for vaccination to meet the requirements of community immunity, HCPs play a vital role in building trust among the population, can see Appendix A. Therefore, a clear and transparent policy with accurate communication with all relevant stakeholders is deemed necessary [16]. Advocacy is another effort applied to influence and impact policy- and decision-makers; to change public perceptions and beliefs; to amend behaviors; and to fight for changes within society. In the efforts to improve immunization and health, advocacy encompasses these definitions in one form or another. Advocacy for immunization includes primary healthcare providers, government officials, and researchers [24].

Vaccination of healthcare providers has recently attracted the attention of public health officials. HCPs have direct contact with patients specifically and with the population in general. In addition, HCPs have a significant influence on the population as people see them as a source of correct and accurate information. Therefore, this study aimed to compare the acceptance rate, advocacy rate, and beliefs about the COVID-19 vaccine among HCPs in two time periods, i.e., November to December 2020 and September to October 2021, which were before and after COVID-19 vaccine approval by authorities.

## 2. Methods

### 2.1. Design

This was a repeated cross-sectional study wherein different HCPs were assessed in two periods ten months apart: November to December 2020 and September to October 2021. It was conducted in Qatif Central Hospital, Eastern Region of Saudi Arabia. The survey was distributed online through emails and WhatsApp to the hospital employees. All healthcare providers in the hospital were targeted (approximately 1000 employees). Employees in Qatif Central include physicians, nurses, dietitians, pharmacists, physiotherapists, radiology technicians, dentists and dental assistants, and paramedics. Eligibility criteria included all HCPs in the above-mentioned hospital. Exclusion criteria specifically for the second period included HCPs who participated in the first-period survey. The survey was sent through the HCPs’ hospital email as well as through WhatsApp. First-period collection survey approval was obtained from the Qatif Central Hospital (QCH-SERCO 270/2021), and the study was conducted in accordance with the Declaration of Helsinki. The study used a non-probability convenience sample, where all respondents from the hospital were enrolled consecutively.

### 2.2. Sample Size

The sample size of the current study was estimated using G*Power3, based on multiple linear regression using the independent two-tailed t-test, with a confidence level of 95%, a margin rate of error at 5%, and a power of 80.0%, with a medium effect size of 0.30. A 10% increase was considered to address the non-responder rate. The minimum required sample size of the current study was 460 subjects: 230 in each study period. However, a total of 609 participants enrolled in the study by completing the distributed study survey: 236 in the first period and 373 in the second period of the study. A total of 1000 online surveys were distributed in each period among hospital healthcare providers, and 609 (60.9%) completed the surveys.

### 2.3. Data Collection Tool and Validation

An online, self-administered, pre-structured questionnaire was used to collect data from the study participants. The survey consisted of two sections; the first section covered the socio-demographic characteristics, and the second section was the Likert scale regarding the HCPs’ perceptions towards COVID-19 vaccinations. The validation test for the survey questions revealed internal consistency of 83%.

### 2.4. Statistical Analysis

In this study, standard statistical procedures were applied using the Statistical Package for Social Sciences (SPSS, version 25). The collected data were validated for accuracy and completeness before conducting the statistical analysis. Descriptive and inferential statistics were conducted, where descriptive analysis was used for socio-demographic information. Frequencies and means ± SD were calculated for categorical and continuous variables, respectively. The Chi-square test was used to compare the two-period groups. A *p*-value of less than 0.05 is considered significant.

## 3. Results

Out of the total HCPs who received the survey, there were 609 responders: 236 participants in the first period and 373 participants in the second period. The two study groups had no statistically significant difference in gender, age, educational degree, years of experience, marital status, and comorbidities. However, there was a significant difference in nationality and occupation between the two study periods.

Female participants were higher in number than male participants for both study periods. Furthermore, around 60% of the study participants of the two different study periods were aged between 26 and 35 years old. More than half of the study participants had a bachelor’s degree, and the study participants were matched for their work experience years. Of the first-period participants, only 16.1% had comorbidities, while 18.8% of the second-period participants had comorbidities. Around three-quarters of the study participants in each period were nurses, whereas physicians and dentists represented around 10% of the study participants (see Table 1).

Out of the 609 participating HCPs in the study, only 13 participants did not get the vaccine. Reasons stated for not having the vaccine were the following: four of them got infected in less than six months, four of them were hesitant to take the vaccine, and five of them were worried about the possible side effects.

Around three-quarters of the first-period study participants agreed to have good medical knowledge regarding vaccine safety and efficacy. However, only 67% of the second-period participants agreed on that. In the first period, 47.88% believed that the COVID-19 vaccines were tested for enough time for safety and efficacy, whereas 40.32% of the study participants believed this in the second period, with a *p*-value of 0.062 between the two study groups. Around half of the participants in the first period believed that the media has a positive impression of the vaccine, and 57.53% of the study participants in the second period believed that. On the other hand, 44.07% and 52.96% of the study participants in the first and second study periods, respectively, believed that honest facts provided to the people about vaccines improve their acceptance. However, 27.54% of the first-period study participants believed that forced vaccination by the government authorities provoke hesitancy, while 44.35% of the second-period study participants believed that; the *p*-value was less than 0.001 (see Table 2).

There was around a 40% difference in the acceptance rate between the two study periods; the latter period was higher at 94.7%. Furthermore, 24.1% was the difference between the willingness to advocate the COVID-19 vaccine for others; the first period had a lower percentage (60.1%). The *p*-value for acceptance rate and willingness for advocacy between the two study periods was <0.001 (see Figure 1).

## 4. Discussion

This is a repeated cross-sectional study on the acceptance, advocacy, and perception among HCPs toward the COVID-19 vaccine. The study questionnaire was distributed twice for different study participants ten months apart between the two study periods to assess the acceptance, advocacy rate, and differences in the COVID-19 vaccine perception within these two time periods. The first period was before issuing the Pfizer-BioNTech and AstraZeneca vaccines, which were both authorized for emergency use in December 2020. However, the second period was conducted after the vaccine approval, in an attempt to assess the difference between pre- and post-vaccine approval.

Evidence from other studies suggested that several HCPs were vaccine-hesitant initially. A qualitative study conducted in four European countries to investigate concerns among HCPs regarding vaccination demonstrated that vaccine hesitancy was present in all four countries among vaccine providers [25]. The result of the previous study is aligned with our study results of the first period, as the acceptance rate was only 55%, indicating some hesitancy [25]. In addition, high trust in health authorities was expressed by the previous study participants, but there was also a strong mistrust of pharmaceutical companies among healthcare providers due to perceived financial interests and lack of information concerning vaccine side effects [25]. Compared to our study, trust in manufacturing countries had an average of 60% in the two-period assessments. In addition, around 55% of the study participants in both periods stated that they trusted the manufacturing companies of the vaccines. Around 15% and 30% in the first and second periods, respectively, believed that the vaccine industries are driven by financial motives rather than health interests.

Linking the two studies, i.e., the study conducted in different European countries and the present study, is hindered by different cultural backgrounds between Saudi and European countries. An interesting study showed that there was a significant association between cultural background and beliefs about the benefits and side effects of medicines among students [26]. Students with Asian backgrounds tended to express more negative views about medications compared to people from European cultural backgrounds [26]. Additionally, Asian cultures were significantly more likely to perceive medicines as harmful [26].

Beyond the above-mentioned reasons for HCPs being vaccine hesitant is autonomy, which was not assessed in this study. Even though HCPs know that they are the bridge for the gap between policymakers and patients, some HCPs believe that their personal choices should not affect patients. A study conducted in Israel showed that nurses did not believe that they should be role models regarding pertussis vaccination, and they have the right to decide whether they want to be vaccinated or not [27].

Surprisingly, 77.12% of the participating HCPs stated that they have good medical knowledge about vaccine safety and efficacy in the first period, while only 67.2% of the participating HCPs agreed on this statement in the second period. The opposite was expected, as the first period had only a little time between the vaccine approval and administration. However, participants had more time to acquire medical knowledge in the second period. HCPs in the first period may have thought that they had good knowledge; however, they might have realized later on that the knowledge they had was more superficial rather than deep enough, as more data came out about vaccines with time. Another explanation is that as two different groups of HCPs were surveyed, the group with more knowledge about vaccine safety and efficacy may have responded in the first period rather than the second period.

In this study, our cohort of HCPs exhibited great progress in trusting the safety of the product and sufficient testing by the manufacturer, with 60% average rates of satisfaction in both study periods. In this regard, several studies have been conducted on issues and challenges associated with vaccine trust and acceptance. While the rapid development of COVID-19 vaccines was a breakthrough for the global population, it is anticipated that vaccines will face many challenges in terms of trust and acceptance. Another study carried out in largely high-income countries mentioned concerns regarding the safety of vaccines against COVID-19 and reported concerns including the rapid pace of vaccine development, as one of the primary reasons for hesitancy [28]. HCPs may have a higher concern with the rapid development of the COVID-19 vaccines, as they are aware of the drug development phases as well as the time needed to test the safety and efficacy of a new investigational product as compared to non-HCPs, who are not aware of the drug or vaccine development process. Trust in the vaccines is vital, and it is critically dependent on the ability of HCPs to first build trust among themselves and then among the community regarding vaccine effectiveness and safety.

The Organization for Economic Co-operation and Development (OECD) Policy Responses to Coronavirus (COVID-19) indicated that only a small minority of the population holds anti-vaccination views and hesitancy regarding COVID-19 vaccination [29]. This resistance can challenge the success of the vaccination campaign, even if there is only a small number of hesitant people. Although HCPs were cited as the most reliable vaccine information source, many HCPs were hesitant to receive vaccines. Studies interviewing a group of professional nurses demonstrated a greater willingness to vaccinate in specific nurses’ categories [30]. These categories include the youngest, the most confident in institutions, and nurses with increased responsibility and higher work stress concerning the management of patients [30].

Studies conducted in the Kingdom of Saudi Arabia revealed that HCPs sharing scientific information about the vaccine in terms of epidemiological details and methodological processes improved the rate of acceptance for the vaccine and optimized the rate of uptake [31]. Additionally, senior health policymakers and leaders who received vaccines in a public display have encouraged more people to accept vaccinations [31]. However, as HCPs are the direct contact between patients and scientific information, 44% and 52.9% of the participating HCPs in the first and second periods, respectively, believed that honest scientific facts about vaccines can provoke acceptance.

A study was conducted in different centers in Riyadh, Saudi Arabia, for healthcare providers about COVID-19 hesitancy between October and November 2020, which was before the approval of the first COVID-19 vaccine [32]. The study showed that only 34.6% of the participating HCPs were willing to receive the vaccine, and only 44% were willing to recommend the vaccine to their patients [32]. However, compared to the second period in our study (which was one more month after the data collection of this study), 55.5% of the participating HCPs were willing to accept the new vaccine, and 60.1% were willing to advocate for the vaccine; both results are higher in percentage compared to the above-mentioned study. However, the sample size for that study was only 159 participants, while it was 236 participants in the first period of our study. Additionally, that study was conducted in three hospitals in Riyadh, which is in the central region of Saudi Arabia, while our study was conducted in Qatif, which is located in the Eastern region of Saudi Arabia. Conducting a study in two different regions of Saudi Arabia can give different results due to cultural and belief differences.

Overall, we observed a great increase in the rate of acceptance and advocacy among HCPs for the vaccine among participants in the second period compared to the first survey’s results. This may be the result of the efforts by the Saudi Ministry of Health, which was trying to educate and teach HCPs the right scientific information based on evidence as well as trying to convince the public through disseminating vaccine information in a convincing scientific way. These findings reflect the fundamental role of HCPs and their potential to influence patient vaccination adoption and improve vaccination confidence.

To our knowledge, this is the first study comparing the acceptance rate, advocacy rate, and perception of COVID-19 vaccines pre- and post-vaccine approval in Saudi Arabia. However, the findings of this study should be interpreted with the limitations of this study. First, the cross-sectional design made it hard to make any association. The second limitation is that the study was conducted only at one hospital in the Eastern region of Saudi Arabia, making it difficult to generalize the results to the whole Saudi population. Third, even though the study was a repeated cross-sectional study in two periods, different HCPs participated in the two periods. Therefore, intra-individual variation was not considered, as we surveyed different people in the two different periods. Fourth, besides the measured and calculated variables that determined vaccine hesitancy in this study, there are some extraneous variables that may not be measured, including income and some personal and family factors. Additionally, the two study groups were not matched in all criteria; there were substantial differences in the nationality and occupation between the two groups.

## 5. Conclusions

Overall, the results of the study showed that vaccine hesitancy, as well as the willingness to advocate for the vaccine, were improved between the pre-vaccine approval period and the post-vaccine approval period, showing the efforts made by the government improved COVID-19 acceptance and advocacy among HCPs. However, vaccine hesitancy is not a new issue but a multifactorial problem. The vital role of HCPs through efficient communication with the community can help build trust. Other stakeholders can develop and provide strategies to educate the population about the benefits and consequences of not receiving the vaccine. For a better understanding of HCPs’ beliefs, a qualitative study is needed.

## Figures and Tables

**Figure 1 vaccines-11-00488-f001:**
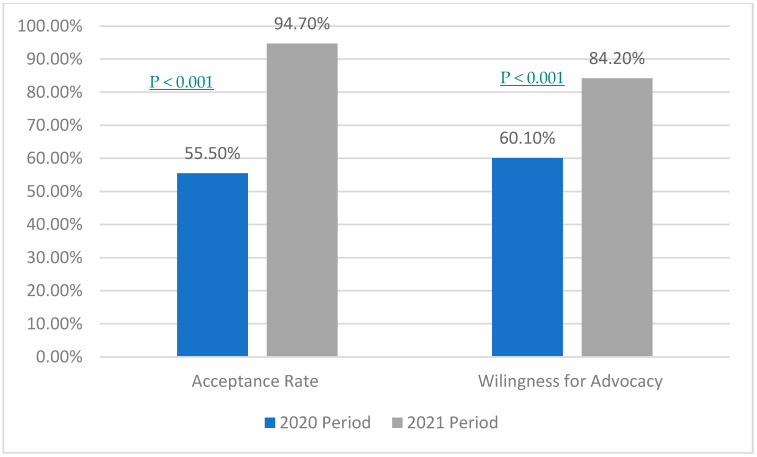
Comparing acceptance and advocacy among health care providers in a 10-month period.

**Table 1 vaccines-11-00488-t001:** Basic socio-demographic characteristics of participants (n = 609).

Characteristics	First Period (236) N (%)	Second Period (373) N (%)	Total (%)	*p*-Value
**Gender** Female Male	208 (88.1) 28 (11.2)	321(86.1) 52 (13.9)	529 (86.9) 80 (13.1)	0.231
**Age group** 25 years and below 26–35 years 36–50 Above 50 years	17 (7.2) 139 (58.9) 72 (30.5) 8 (3.4)	15 (4) 225 (60.3) 122 (32.7) 11 (3)	32 (5.3) 364 (59.8) 194 (31.9) 19 (3.1)	0.369
**Educational degree** Bachelor Diploma Master Degree PhD Postgraduate Diploma	143 (60.6) 59 (25) 16 (6.8) 9 (3.8) 9 (3.8)	241 (64.6) 87 (23.3) 15 (4.1) 21 (5.6) 9 (2.4)	384 (63.1) 146 (24.0) 31 (5.1) 30 (4.9) 18 (3.0)	0.327
**Nationality** Saudi Non-Saudi	205 (86.9) 31 (13.1)	265 (71.1) 108 (28.9)	470 (77.2) 139 (22.8)	<0.001
**Total years of experience** 5 years and less 6–10 years More than 10 years	80 (33.9) 62 (26.3) 94 (39.8)	121 (32.4) 100 (26.8) 152 (40.8)	201 (33.0) 162 (26.6) 246 (40.4)	0.932
**Marital status** Married Single Divorced Widowed	187 (79.2) 46 (19.5) 3 (1.3) 0 (0)	288 (77.2) 74 (19.8) 10 (2.7) 1 (0.3)	475 (78.0) 120 (19.7) 13 (2.1) 1 (0.2)	0.674
**Have comorbidity** Yes No	38 (16.1) 198 (83.9)	70 (18.8) 303 (81.2)	108 (17.7) 501 (82.3)	0.401
**Occupation** Physicians and dentists Pharmacist Nurses Other HCPs	22 (9.3) 7 (3) 178 (75.4) 29 (12.3)	52 (13.9) 3 (0.8) 286 (76.7) 32 (8.6)	74 (12.15) 10 (1.64) 464 (76.2) 61 (10)	0.034

**Table 2 vaccines-11-00488-t002:** Thoughts and beliefs of HCPs regarding COVID-19 vaccines.

	First Period (236) N (%)	Second Period (373) N (%)	*p*-Value
Agreed to have good medical knowledge of vaccine safety and efficacy	182 (77.12)	250 (67.2)	0.007
Believed vaccines are tested long enough for safety and efficacy	113 (47.88)	150 (40.32)	0.062
Trusted the manufacturing country of the vaccine	150 (63.56)	218 (58.6)	0.208
Trusted the manufacturing company of the vaccine	133 (56.36)	204 (54.84)	0.687
Believed the vaccine industry is driven by financial motives and not health interest	41 (17.37)	118 (31.72)	<0.001
Believed media has created a positive impression of the vaccine	111 (47.03)	214 (57.53)	0.012
Believed honest scientific facts on vaccines provoke acceptance	104 (44.07)	197 (52.96)	0.035
Believed forced vaccination by authorities provokes hesitancy	65 (27.54)	165 (44.35)	<0.001

## Data Availability

All authors have read and approved the manuscript for publication. Availability of data and material: Data used and analyzed in this study will be promptly available from the publisher upon request.

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
