# Peer review of "Acceptance, Advocacy, and Perception of Health Care Providers on COVID-19 Vaccine: Comparing Early Stage of COVID-19 Vaccination with Latter Stage in the Eastern Region of Saudi Arabia"

_vaccines, 2023, doi:10.3390/vaccines11020488_

Round 1

Reviewer 1 Report

The level of critical insight in the discussion could be improved by offering alternative explanations/theory/evidence as to why HCPs might be vaccine 'hesitant' beyond those few factors included ghere. Limitations must include discussion around Extraneous variables that could help to understand why HCP's might be hesitant to have a an experimental mRNA injection that is yet to undergo long-term efficacy/safety trials. No alternative perspectives are being considered in this paper, hence this paper provides a skewed perspective on the subject.

The discussion could also offer a greater awareness of the possibility that HCPs are potentially more attuned to some of the genuine concerns around this vaccine; and understanding cannot be compared to other estabilshed vaccines that have undergone rogorous testing. These points have not been raised in the discussion, but warrant consideration to help balance-out the overall message that this paper puts across.

Reviewer 2 Report

Dear, Authors 

I read your manuscript " Acceptance, Advocacy and Perception of Health Care Providers on COVID-19 Vaccine: Repeated Cross-sectional Study in Eastern Region of Saudi Arabi" with interest. 

I do have some suggestions for your kind consideration. 

Title:

You have compared pre and post-vaccination avalibility data; that is a good selling point and you should include in the title somehow. That will attract more citations of the work in the future. Just a suggestion 

1. background:

"Although several efforts are underway to explore treatment options, currently, there has been no proven treatment available which adds to the growing concern"  - This statement is not true : steroids, remdisivir , paxilovid, immunomodulators: Also vaccination are available. Please modify the statement and reference to reflect this one.  

Methods: 

Was there any specific age group you included or excluded? If yes, please mention it here. 

Also, please define who is included in HCPs here. Like doctors, nurse or everyone. 

Sample size : 

When the authors mentioned that 1000 online surveys were distributed, I am guessing they emailed it to 1000 participants. Please clarify how HCPs were recruited. 

Data collection tool : 

Which language was the survey done and which language do most HCPs speak? Or please mention what is the native language in the hospital ?. The language in the survey was done can cause selection bias if not considered carefully. It seems like there is a good amount of participants who were not Saudi nationals

Please add your survey tool to the supplementary material for the readers to review. 

Results : 

Use a similar way of presenting the data, like if you use percentages, then use percentages throughout. Authors have used somewhere percentages and somewhere fractions 

The authors have done simple descriptive stats in the two groups. I am not sure if the authors considered at least comparing the two groups and making sure there is no significant difference in the two groups. This might increase the validity of simple stats. Also, would authors be able to show the difference in simple descriptive stats are significant or not? Please do the stats again

Discussion:

A study conducted in Europe consisting of 65 semi[1]structured interviews with vaccine providers in Croatia, France, Greece, and Romania investigated concerns among HCPs regarding vaccination, results demonstrated that vaccine hesitancy was present in all four countries among vaccine providers”   --  Vaccine providers ? or patients ? Please calrify

Paragraph 2 nd – Authors talked to much about reference 25, rather than their findings. Sometimes, it is confusing to the reviewer whether they are talking about their data or reference number 25 data. – Please improve this paragraph. Also I am not sure what the rationale is for mentioning reference number 26 regarding their data or reference number 25 ? can the authors please explain?

 3 paragraph:  - Can authors explain possible explanations of these unexpected findings? Readers can look at table 2 and make this inference, but it would be interesting to see what authors think caused these exciting findings. 

9 paragraph - 

" To our knowledge, this is the first study comparing the acceptance rate, advocacy rate, and perception about COVID-19 vaccines pre and postvaccine approval." I am guessing authors means to say saudi arabia ? Please calrify 

Conclusion : 

" Overall results of the study showed that efforts done by the government improved COVID-19 acceptance and advocacy among HCPs" 

what efforst were made by goverment . please put reference or description. 

Reviewer 3 Report

Estimated Authors,

I've read with great interest your original article entitled "Acceptance, Advocacy and Perception of Health Care Providers on COVID-19 Vaccine: Repeated Cross-sectional Study in Eastern Region of Saudi Arabia".

In this cross-sectional study, through a 2-time sampling technique (with a first round performed in an early stage of vaccination campaign, and a second round more lately), Authors were able to characterize and address some features of the vaccine acceptance among healthcare providers from the Eastern Regions of the KSA.

This article, albeit it could be hardly considered radically original, shares some interesting points over the overall barriers / motivators of SARS-CoV-2 vaccines among HCP, and could be therefore considered of certain interest for the aims and the scope of the readers of Vaccines.

Still, I've some concerns about the overall data reporting that should be carefully addressed before the eventual acceptance of this study.

More precisely:

1) Authors must provide some information about the targeted population and the settings of the recruitment of study participants. How many HCW where originally targeted? how many of them had replied to the original invitation? how was the invitation shared? If the questionnaire was originally delivered through (for example) a internet based platform, some degree of self selection of participants could be suspected (see for example the comments from this study: https://pubmed.ncbi.nlm.nih.gov/34452014/).

2) in order to complement the point 1), please include a flowchart of study participants;

3) Table 1 should be improved by a statistical analysis, at least an univariate one, providing some basic comparisons between first and second stage of the analysis, in order to understand whether the study participants were or were not comparable in the two stages of the study;

4) Similar considerations have to be shared with data included in Table 2.

5) I would have expected that authors had explored the factors associated with at least one of the two statements included in Figure 1, at least as an univariate analysis (it would be more accurate and interesting the implementation of a multivariable analysis taking in account either acceptance rate or willingness for advocacy, or both, and demographics + thoughts and beliefs as explanatory variables; again see the aforementioned example for some modelization of a potential design approach).

Round 2

Reviewer 1 Report

The paper requires proof reading as there are multiple grammatical and typographical errors.

The revisions require empirical evidence with citations added to support the additional points made in the discussion.

Reviewer 3 Report

Estimated Authors,

I've appreciated the clear and extensive efforts you paid in order to improve your paper (as you did). The paper has been radically improved, and I think that, now, it is only a small step away from the final acceptance.

More precisely, I think that the improved analyses you had performed stress the opportunity for improving the LIMITS section of this study.

In fact, two issues now shine very clearly:

1) the substantial differences between the sample of T1 and T2 from a demographic point of view (i.e. individuals of Saudi Nationality dropped from 86.9% to 71.1%, the share of Physicians increased from 9.3% to 13.9%)

2) the selection of the participants: while the recruitment procedures were quite conventional (when dealing with internet based questionnaires, at least), the potential shortcoming represented by the identification of the individual and the attention deserved to the privacy and confidentiality of shared information are limitedly treated. Authors should discuss whether the procedure of recruitment and the subsequent analysis of questionnaire may have led the participants towards some sort of "social desirability bias".
